# Hyperbolic Implicit Equilibrium

## Abstract

Euclidean geometry has long dominated neural networks and deep learning, yet neuroscience reveals a different picture. At the representational level, spatial and mnemonic maps in the brain are naturally organized in hyperbolic geometry, supporting efficient hierarchical embeddings. Hyperbolic neural networks exploit this property but remain shallow and costly: explicit architectures must retain all activations, and curvature-induced distortions make stability difficult, leading to prohibitive memory and runtime overhead. At the dynamical level, neural activity tends to converge to stable equilibrium states, conferring robustness, stability, and energy efficiency. Motivated by these complementary principles, we establish Hyperbolic Implicit Equilibrium (HIE), the first implicit equilibrium framework for hyperbolic networks. HIE directly solves for a fixed point and trains via implicit differentiation, requiring only a single Jacobian–vector product. This design enables models of effectively infinite depth within a constant memory footprint, while hyperbolic contraction accelerates convergence beyond Euclidean counterparts. We further contribute Lorentz group normalization for stable equilibrium and a complete theoretical analysis of optimization, stability, and generalization. Experiments show that HIE scales hyperbolic models far beyond prior explicit designs, achieving faster and more robust convergence and revealing the unique benefits of hyperbolic geometry for implicit deep learning.

## 1 Introduction

Two thousand years ago, Euclid postulated that through a point not on a line there exists exactly one line parallel to it—a principle that shaped the geometry of human thought for centuries. Yet our brains appear to think differently: neuroscientific evidence (Zhou et al., 2018a; Zhang et al., 2023; Longhena et al., 2025) reveals that neural representations of space and memory naturally form in hyperbolic geometry, where infinitely many parallels can pass through such a point, geodesics diverge rapidly, and volume grows exponentially, enabling hierarchies to be embedded with remarkable efficiency. This divergence between the geometry of intuition and the geometry of cognition is striking. It suggests that if our goal in artificial intelligence is to build systems that learn and reason as effectively and efficiently as the brain, then continuing to rely exclusively on Euclidean geometry may be fundamentally limiting. Hyperbolic neural networks (HNN) have therefore emerged as a pathway to bridge this gap, extending deep learning into negatively curved spaces.

HNN promises compact and faithful representations of inherently structured data, a property less naturally captured in Euclidean space. This is evident in vision and language, where taxonomies in ImageNet, part–whole relations in object recognition, and semantic trees in language all exhibit hierarchies that hyperbolic geometry can embed with exponentially low distortion (Khrulkov et al., 2020). However, scaling HNN to the level required for modern tasks remains profoundly challenging. Curvature-induced volume growth and nonlinear gyrovector operations complicate optimization; explicit architectures must retain intermediate activations to preserve manifold consistency, leading to prohibitive memory costs. As depth increases, distortions accumulate and training becomes numerically fragile. As a result, most hyperbolic models remain shallow. This scalability bottleneck is a structural consequence of the interplay between hyperbolic geometry and optimization dynamics, limiting the expressive power of hyperbolic representations on large-scale datasets.

Several attempts have been made to alleviate these challenges. Poincaré ResNets (Van Spengler et al., 2023) derived custom backward operators to shrink the computational graph, reducing memory at the cost of higher computational complexity. Later works introduced feature clipping and

Euclidean reparameterizations (Mishne et al., 2023; Guo et al., 2022; Mathieu et al., 2019), which improved numerical stability and enabled training with 32-bit rather than 64-bit precision (Bdeir et al., 2024), indirectly lowering memory usage. While such techniques improved local stability, the fundamental issue remains: layers are still unrolled sequentially, intermediate activations must be stored, and the compounded cost of hyperbolic operations quickly overwhelms memory and runtime budgets. As noted in Poincaré ResNet (Van Spengler et al., 2023) and HyperbolicCV (Bdeir et al., 2024), the memory footprint of HNN substantially exceeds Euclidean networks. The core tension persists: the same geometry that enables compact hierarchical representations also renders explicit architectures unstable and computationally heavy, leaving the scalability problem unresolved.

To overcome this long-standing barrier, we take a different perspective inspired by the brain itself. Recent neuroscientific studies (Englert et al., 2024; Song et al., 2024) have shown that neural activity often converges to stable attractor states, ensuring efficiency in energy use and memory retrieval. Analogously, we establish Hyperbolic Implicit Equilibrium (HIE), which formalizes the deep equilibrium (DEQ) (Bai et al., 2019; 2020) framework in hyperbolic networks. Instead of propagating through layers explicitly, HIE directly solves for a fixed-point representation using a black-box equilibrium solver. Gradients are computed via implicit differentiation, requiring only a single Jacobian–vector product at equilibrium and bypassing layer-wise backpropagation. This design enables constant-memory training, reduces runtime, and crucially allows hyperbolic models to scale to large datasets without sacrificing efficiency. Moreover, HIE requires fewer iterations than Euclidean counterparts, yielding faster convergence and improved stability. Finally, we establish a complete theoretical analysis of optimization, stability, and generalization that explains why HIE is not only trainable but also more stable, efficient, and generalizable than both explicit HNN and Euclidean DEQ. Our main contributions are as follows:

1. We establish the implicit equilibrium framework for hyperbolic neural networks, grounded in two principles of brain: hyperbolic representations and equilibrium dynamics;
2. We design Lorentz group normalization, a geometry-preserving normalization method that maintains manifold consistency while improving the stability of equilibrium solvers;
3. We present a comprehensive theoretical analysis of optimization, stability, and generalization, which establishes rigorous convergence and robustness guarantees;
4. We show that HIE unlocks deep hyperbolic models with constant memory and accelerated training, overcoming the scalability bottleneck of prior hyperbolic neural networks;
5. We further demonstrate that HIE converges faster to fixed points than Euclidean implicit equilibrium models, leveraging the unique benefits of hyperbolic geometry.

## 2 BACKGROUND

### 2.1 HYPERBOLIC NEURAL NETWORKS

Hyperbolic neural networks (HNN) leverage negatively curved spaces to compactly represent hierarchical structures that are distorted in Euclidean embeddings (Nickel & Kiela, 2018; Ganea et al., 2018; Peng et al., 2021). Initially, Euclidean encoders projected features into hyperbolic heads for classification (Khrulkov et al., 2020; Liu et al., 2020; Guo et al., 2022), segmentation (Hsu et al., 2021; Atigh et al., 2022), generation (Mathieu et al., 2019; Nagano et al., 2019; Ovinnikov, 2019), and metric learning (Yan et al., 2021; Ermolov et al., 2022; Yue et al., 2023). Further, hyperbolic formulations generalized fundamental layers—convolutions, attention, and normalization—either in the Poincaré ball (Ganea et al., 2018; Shimizu et al., 2020; Van Spengler et al., 2023) or the Lorentz model (Chen et al., 2021; Fan et al., 2022). Recently, HyperbolicCV (Bdeir et al., 2024) introduced Lorentz CNN, providing missing components such as hyperbolic convolution, batch normalization, and logistic regression.

Nevertheless, hyperbolic networks remain computationally heavy. The exponential volume growth of the Lorentz model causes instability; remedies such as feature clipping and Euclidean reparameterizations (Mishne et al., 2023; Guo et al., 2022; Mathieu et al., 2019) only improve local stability. Poincaré ResNets (Van Spengler et al., 2023) reduced memory by custom backward operators, but at higher runtime cost. Most prior work has therefore centered on stability rather than fundamentally addressing the scalability problem.

## 2.2 IMPLICIT DEEP LEARNING

Implicit deep learning departs from explicit multi-layer architectures by defining hidden states through analytical conditions rather than prescribed computation graphs. This line of work dates back to implicit differentiation for recurrent dynamics (Pineda, 1987; Almeida, 1990), later revisited as recurrent back-propagation (RBP) (Liao et al., 2018). Recent interest has revived implicit formulations across diverse domains (El Ghaoui et al., 2021; Gould et al., 2021). Representative examples include Neural ODE (NODE) (Chen et al., 2018; Dupont et al., 2019), which interpret residual networks as continuous dynamics solved by ODE integrators. Other instantiations range from optimization-based layers (Amos & Kolter, 2017; Djolonga & Krause, 2017), differentiable physics simulators (de Avila Belbute-Peres et al., 2018; Qiao et al., 2020), and logical reasoning modules (Wang et al., 2019), to continuous-time generative models (Grathwohl et al., 2018).

Deep equilibrium models (DEQ) (Bai et al., 2019) solve for fixed points via black-box root-finding, achieving representations equivalent to infinitely deep weight-tied networks while requiring only $O(1)$ memory through implicit differentiation. Multiscale DEQ (MDEQ) (Bai et al., 2020) extended the approach to vision, jointly solving for equilibrium across resolutions and enabling classification and segmentation within one model. Despite these advantages, DEQ exhibit two main drawbacks: (i) convergence relies heavily on contractive dynamics, often unmet in practice, and (ii) runtime scales with the number of root-finding iterations. As noted in MDEQ, iterations are truncated at a threshold, yielding only approximate equilibrium. Our contribution builds on this line by establishing hyperbolic implicit equilibrium (HIE), where negative curvature strengthens contraction, accelerates solver convergence, and thus reduces runtime while preserving constant memory.

## 3 METHOD

### 3.1 PRELIMINARIES

**Hyperbolic Geometry.** Hyperbolic space is a prototypical example of a Riemannian manifold with constant negative curvature (Cannon et al., 1997; Ratcliffe, 2006). Formally, the $n$-dimensional hyperbolic space $\mathbb{H}_K^n$ can be described as a pair $(\mathcal{M}^n, g^K)$, where $\mathcal{M}^n$ denotes the underlying manifold and $g^K$ the Riemannian metric associated with curvature $K < 0$. Several equivalent models exist to represent hyperbolic geometry, such as the Poincaré ball and the Lorentz hyperboloid. In this work, we adopt the Lorentz model due to its favorable numerical behavior and the availability of simple closed-form operations, including exponential and logarithmic maps, geodesic distance, and parallel transport. These properties make it particularly suitable for integration with deep learning methods. For completeness, detailed derivations of these operators are provided in Appendix A.

**Lorentz Model.** The Lorentz model offers a convenient representation of hyperbolic geometry with curvature $K < 0$, as shown in Figure 1. Let $\mathbb{R}^{d+1}$ be equipped with the Lorentzian bilinear form

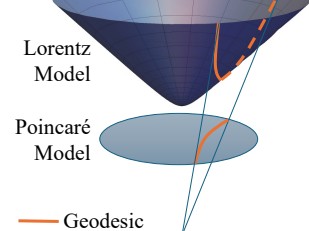

$$\langle \mathbf{u}, \mathbf{v} \rangle_{\mathcal{L}} = -u_0 v_0 + \sum_{i=1}^{d} u_i v_i, \quad (1)$$

where $\mathbf{u} = (u_0, \ldots, u_d), \mathbf{v} = (v_0, \ldots, v_d) \in \mathbb{R}^{d+1}$. The $d$-dimensional hyperbolic space of curvature $K$ can then be realized as the upper sheet of the two-sheeted hyperboloid

$$\mathbb{H}_K^d = \left\{ \mathbf{u} \in \mathbb{R}^{d+1} \,\middle|\, \langle \mathbf{u}, \mathbf{u} \rangle_{\mathcal{L}} = \tfrac{1}{K}, \ u_0 > 0 \right\}. \quad (2)$$

Figure 1: Visualization of Lorentz and Poincaré model.

The geodesic distance between two points $\mathbf{u}, \mathbf{v} \in \mathbb{H}_K^d$ is given by

$$d_{\mathbb{H}_K}(\mathbf{u}, \mathbf{v}) = \frac{1}{\sqrt{-K}} \operatorname{arcosh}\big(K \cdot (-\langle \mathbf{u}, \mathbf{v} \rangle_{\mathcal{L}})\big). \quad (3)$$

For simplicity and consistency with prior work in hyperbolic deep learning, we use curvature $K < 0$ and set $K = -1$ for all experiments. More notations and conventions are provided in Appendix A.

## 3.2 Hybrid Network Architecture

Following recent work (Bdeir et al., 2024), we adopt a hybrid architecture that integrates Euclidean and hyperbolic components. The encoder combines Euclidean and Lorentzian residual blocks, while the task head is fully hyperbolic. This division balances efficiency and expressivity: Euclidean layers provide cost-effective feature extraction, whereas hyperbolic modules capture hierarchical relations. Empirical studies show that intermediate representations exhibit varying degrees of $\delta$-hyperbolicity (Khrulkov et al., 2020), suggesting that only certain layers benefit from negative curvature. By introducing hyperbolic operators selectively, the model leverages their representational power while avoiding the prohibitive memory and runtime overhead of fully hyperbolic encoders.

To predict when hyperbolic equilibrium models are advantageous, we employ the notion of $\delta$-hyperbolicity (Khrulkov et al., 2020), which quantifies the tree-likeness of a metric space. In practice, we estimate $\delta$ from encoder features and compute the normalized score $\delta_{\mathrm{rel}}$ across scales. Hyperbolic operators are used when representations exhibit strong hyperbolicity, and Euclidean ones otherwise. The computation details are provided in Appendix B.1.

## 3.3 Hyperbolic Residual Block

Our network integrates several hyperbolic operators into residual blocks. Specifically, we employ Lorentz convolutional layers, a Lorentz multinomial logistic regression (MLR) classifier, and hyperbolic residual connections with non-linear activations. These modules extend their Euclidean counterparts while preserving consistency with the Lorentz model. Convolutions and classifiers operate directly in hyperbolic space, while residual connections and activations are defined via exponential and logarithmic maps between the tangent space and the manifold. Together, they form the building blocks of our hybrid architecture, enabling expressive yet stable learning within curved geometry. Implementation details and closed-form formulas are deferred to Appendix B.2.

**Lorentz Group Normalization.** Normalization is indispensable in equilibrium models: batch-dependent methods such as BatchNorm are ill-suited, as they inflate the Jacobian norm of $f_\theta$ and destabilize implicit solvers (Bai et al., 2019; 2020). MDEQ replaces BatchNorm with Group-Norm (Wu & He, 2018), which normalizes feature subsets within each sample, removing batch-size dependence and improving solver stability. However, directly applying Euclidean GroupNorm to hyperbolic layers is invalid: channel-wise statistics computed in $\mathbb{R}^d$ break manifold constraints and lead to drift off the Lorentz hyperboloid.

To address this, we design *Lorentz Group Normalization (LGN)*, which extends GroupNorm to the Lorentz model $\mathbb{H}_K^d$ while preserving geometric consistency. LGN operates on groups of Lorentz points (e.g., all spatial positions assigned to a feature group within a sample). Design notes and variants are provided in Appendix B.3. For group $g$ with points $S_g = \sum_{j=1}^{N_g} \mathbf{u}_j$, we first compute the Lorentz mean

$$\boldsymbol{\mu}_g = \frac{S_g}{\sqrt{K \langle S_g, S_g \rangle_{\mathcal{L}}}}. \tag{4}$$

which lies on $\mathbb{H}_K^d$ and minimizes the Fréchet energy. We estimate dispersion by the group variance

$$\sigma_g^2 = \frac{1}{N_g} \sum_{j=1}^{N_g} d_{\mathbb{H}_K}(\mathbf{u}_j, \boldsymbol{\mu}_g)^2. \tag{5}$$

where $d_{\mathbb{H}_K}$ is the hyperbolic distance in the Lorentz model.

Each point $\mathbf{u}_j$ is mapped to the tangent space at $\boldsymbol{\mu}_g$ by $\log_{\boldsymbol{\mu}_g}^K$, parallel transported (PT) to the origin $\mathbf{o}$, rescaled by $\frac{\gamma_g}{\sqrt{\sigma_g^2 + \varepsilon}}$, and transported to a learnable anchor $\boldsymbol{\beta}_g \in \mathbb{H}_K^d$ before mapping back via $\exp_{\boldsymbol{\beta}_g}^K$, which is formulated as

$$\mathrm{LGN}_g(\mathbf{u}_j) = \exp_{\boldsymbol{\beta}_g}^K \left( \mathrm{PT}_{\mathbf{o} \to \boldsymbol{\beta}_g}^K \left( \frac{\gamma_g}{\sqrt{\sigma_g^2 + \varepsilon}} \, \mathrm{PT}_{\boldsymbol{\mu}_g \to \mathbf{o}}^K \left( \log_{\boldsymbol{\mu}_g}^K (\mathbf{u}_j) \right) \right) \right). \tag{6}$$

Here $\gamma_g$ is a learnable scale and $\boldsymbol{\beta}_g$ a learnable bias. All steps—exponential/logarithmic maps, distances, and parallel transport—have closed forms in the Lorentz model, ensuring numerical stability.

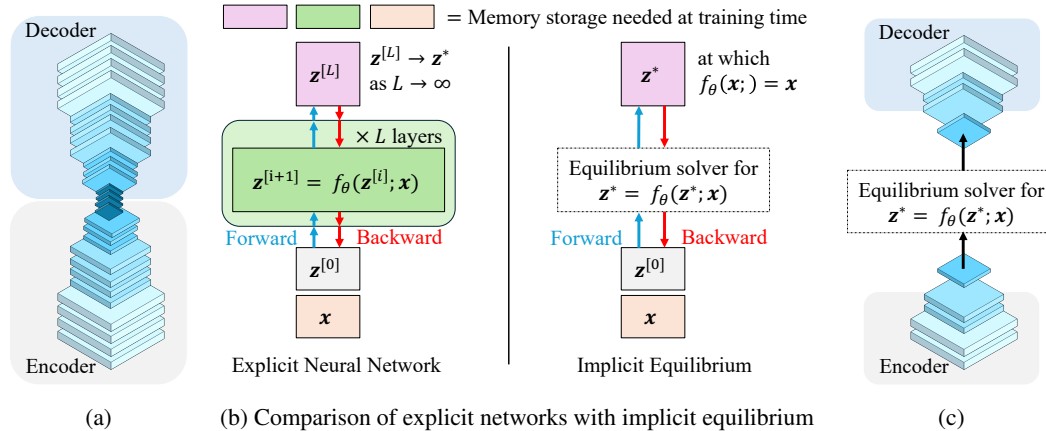

(a)     (b) Comparison of explicit networks with implicit equilibrium     (c)

Figure 2: (a) Explicit network. (b) Memory cost: explicit vs. implicit. (c) Deep equilibrium model.

### 3.4 MULTISCALE IMPLICIT EQUILIBRIUM

Traditional explicit architectures such as ResNet (Figure 2a) unroll a fixed sequence of $L$ layers, requiring storage of all intermediate activations during training. Deep equilibrium models (Figure 2c) replace this explicit stacking with a fixed-point formulation, directly solving for the equilibrium state $z^\star = f_\theta(z^\star; x)$. As summarized in Figure 2b, this implicit formulation reduces memory consumption from $\mathcal{O}(L)$ to $\mathcal{O}(1)$, since intermediate states need not be stored and gradients can be computed through implicit differentiation.

Building on these principles, we adopt a multiscale equilibrium construction inspired by MDEQ (Bai et al., 2020). Formally, let $z = [z_1, \ldots, z_n]$ denote the collection of hidden states at $n$ resolutions, with $z_i \in \mathbb{R}^{H_i \times W_i \times C_i}$. Each resolution evolves through a Lorentz residual block followed by cross-scale fusion, producing updated features $\tilde{z}_i$. As shown in Figure 3, the joint transformation $f_\theta(z; x)$ then acts on all scales, and the equilibrium state is defined as the solution of

$$z^\star = f_\theta(z^\star; x), \qquad (7)$$

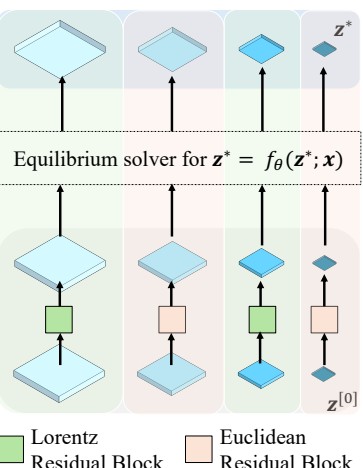

Figure 3: Our hyperbolic implicit equilibrium framework.

where $x$ denotes the input injected at the highest resolution.
This equilibrium is solved simultaneously across scales using a limited-memory quasi-Newton method (e.g., Broyden updates), yielding synchronized hidden states $\{z_i^\star\}_{i=1}^n$. Details of the forward and backward passes of the implicit equilibrium are provided in Appendix C.

## 4 THEORY

**Lemma 1** (Bi-Lipschitz bounds of $\exp^K$ and $\log^K$ on a geodesic ball). *Let $K < 0$, $\kappa = \sqrt{-K}$, and fix $\beta \in \mathbb{H}_K^d$. For $B_R(\beta) = \{z \in \mathbb{H}_K^d : d_{\mathbb{H}_K}(z, \beta) \leq R\}$,*

$$L_{\exp}(R) = \sup_{\|v\| \leq R} \left\| d(\exp_\beta^K)_v \right\| = \frac{\sinh(\kappa R)}{\kappa R}, \qquad L_{\log}(R) = \sup_{z \in B_R(\beta)} \left\| d(\log_\beta^K)_z \right\| = \frac{\kappa R}{\sinh(\kappa R)}.$$

*Hence $L_{\exp}(R) \geq 1$, $L_{\log}(R) \leq 1$, and $L_{\exp}(R) L_{\log}(R) = 1$.*

*Proof sketch.* In constant curvature spaces, the singular values of $d\exp_\beta$ along radial/tangential directions are governed by Jacobi fields, yielding $\sinh(\kappa r)/(\kappa r)$ at geodesic radius $r$; invertibility gives the $d\log_\beta$ bound and the product identity. Full proof in Appendix D.1. $\qquad \square$

**Theorem 1** (Contraction with geodesic damping and radial shrinkage). *Let $K < 0$, $\kappa = \sqrt{-K}$, fix $\beta \in \mathbb{H}_K^d$ and $R > 0$. Define $F : B_R(\beta) \to \mathbb{H}_K^d$ by*

$$F(z) = \exp_\beta^K\Big(\tau A(S(u)) + (1-\tau)u + b(x)\Big), \qquad u = \log_\beta^K(z), \tag{8}$$

*where $0 < \tau \le 1$, $A : T_\beta\mathbb{H}_K^d \to T_\beta\mathbb{H}_K^d$ is linear with $\|A\|_2 \le \alpha$, $b(x) \in T_\beta\mathbb{H}_K^d$, and $S(u) = \psi(\|u\|)\, u/\|u\|$ (with $S(0) = 0$) is $\eta$-Lipschitz on $\{\|u\| \le R\}$ for some $0 \le \eta < 1$. Assume $F(B_R(\beta)) \subseteq B_R(\beta)$. Then on $(B_R(\beta), d_{\mathbb{H}_K})$,*

$$\mathrm{Lip}(F) \;\le\; L_{\exp}(R)\big(\tau\alpha\eta + (1-\tau)\big) L_{\log}(R) \;=\; \tau\alpha\eta + (1-\tau) \;:=\; \rho_\mathbb{H} < 1. \tag{9}$$

*Thus $F$ has a unique fixed point $z^\star \in B_R(\beta)$ and Picard iteration converges linearly with rate $\le \rho_\mathbb{H}$.*

*Proof sketch.* For $z_i \in B_R(\beta)$, write $u_i = \log_\beta(z_i)$ and $\Phi(u) = \tau A(S(u)) + (1-\tau)u + b(x)$. Lemma 1 gives $d_{\mathbb{H}_K}(F(z_1), F(z_2)) \le L_{\exp}(R)\|\Phi(u_1) - \Phi(u_2)\|$ and $\|u_1 - u_2\| \le L_{\log}(R)\, d_{\mathbb{H}_K}(z_1, z_2)$. Using $\|A\| \le \alpha$ and $S$ being $\eta$-Lipschitz yields equation 9. Banach fixed-point theorem concludes. Full proof in Appendix D.2. $\square$

**Corollary 1** (Hyperbolic vs. Euclidean rate). *For $F_\mathbb{R}(u) = \tau Au + (1-\tau)u + b(x)$ on $T_\beta \cong \mathbb{R}^d$, $\rho_\mathbb{R} = \tau\alpha + (1-\tau)$. Under Theorem 1, $\rho_\mathbb{H} = \tau\alpha\eta + (1-\tau) \le \rho_\mathbb{R}$, with strict inequality if $\tau\alpha > 0$ and $\eta < 1$.*

*Proof sketch.* Immediate from $\eta < 1$ and the expressions of $\rho_\mathbb{H}$ and $\rho_\mathbb{R}$. Full proof in Appendix D.3. $\square$

**Theorem 2** (Bounded implicit gradients under contraction). *Let $z^\star$ be the unique fixed point of $F_\theta$ in Theorem 1. All Jacobians and operator/vector norms are taken in $T_{z^\star}\mathbb{H}_K^d$ with metric $g_{z^\star}$. For any differentiable loss $\mathcal{L}(z^\star)$,*

$$\big\|\nabla_\theta\mathcal{L}\big\| \;\le\; \big\|(I - D_z F_\theta(z^\star; x))^{-1}\big\| \,\big\|\partial_\theta F_\theta(z^\star; x)\big\| \,\big\|\nabla_z\mathcal{L}(z^\star)\big\|$$

$$\le\; \frac{1}{1 - \rho_\mathbb{H}}\,\big\|\partial_\theta F_\theta(z^\star; x)\big\|\,\big\|\nabla_z\mathcal{L}(z^\star)\big\|. \tag{10}$$

*Proof sketch.* Differentiate $F_\theta(z^\star; x) = z^\star$ to get $(I - D_z F_\theta)\, dz^\star = (\partial_\theta F_\theta)\, d\theta$; solve via Neumann series since $\|D_z F_\theta\| \le \rho_\mathbb{H} < 1$. Take norms in $T_{z^\star}$ to obtain equation 10. Full proof in Appendix D.4. $\square$

**Lemma 2** (Hyperbolic law of cosines). *Let $\beta \in \mathbb{H}_K^d$, $\kappa = \sqrt{-K}$. For $u, v \in \mathbb{H}_K^d$ with $r_i = d_{\mathbb{H}_K}(u_i, \beta)$ and tangent angle $\theta \in [0, \pi]$ between $\log_\beta^K(u)$ and $\log_\beta^K(v)$,*

$$\cosh\big(\kappa\, d_{\mathbb{H}_K}(u, v)\big) = \cosh(\kappa r_1)\cosh(\kappa r_2) - \sinh(\kappa r_1)\sinh(\kappa r_2)\cos\theta.$$

*Proof sketch.* Standard form in the Lorentz/Poincaré models via geodesic polar coordinates. Full proof in Appendix D.5. $\square$

**Theorem 3** (Geodesic margin and Euclidean comparison). *Assume class means $\mu_c \in \mathbb{H}_K^d$ share radius $r = d_{\mathbb{H}_K}(\mu_c, \beta)$ and $\theta_0 = \min_{c \ne c'} \angle(\log_\beta^K(\mu_c), \log_\beta^K(\mu_{c'}))$. Then*

$$\gamma_\mathbb{H} := \min_{c \ne c'} d_{\mathbb{H}_K}(\mu_c, \mu_{c'}) = \frac{2}{\kappa}\, \mathrm{arcsinh}\Big(\sinh(\kappa r)\sin\tfrac{\theta_0}{2}\Big) \;\ge\; 2r\sin\tfrac{\theta_0}{2} \;=:\; \gamma_\mathbb{R}, \tag{11}$$

*with strict inequality if $r > 0$ and $\theta_0 \in (0, \pi)$.*

*Proof sketch.* Apply Lemma 2 with $r_1 = r_2 = r$, use $\cos\theta = 1 - 2\sin^2(\theta/2)$ to obtain the closed form. Since $\sinh(\kappa r) \ge \kappa r$ and $\mathrm{arcsinh}$ is increasing and concave, $\mathrm{arcsinh}(\sinh(\kappa r)s) \ge s\,\kappa r$ for $s = \sin(\theta_0/2)$, yielding $\gamma_\mathbb{H} \ge \gamma_\mathbb{R}$. Full proof in Appendix D.6. $\square$

Table 1: The hyperbolicity values $\delta_{rel}$ calculated for intermediate embeddings on different datasets.

| Dataset | CIFAR-10 | | ImageNet | | Cityscapes | |
|---|---|---|---|---|---|---|
| Model | MDEQ-small | MDEQ-large | MDEQ-small | MDEQ-XL | MDEQ-small | MDEQ-XL |
| Initial Conv. | 0.2526 | 0.2911 | 0.2230 | 0.2790 | 0.3726 | 0.3599 |
| Branch 1 | 0.1928 | 0.2021 | 0.2022 | 0.1836 | 0.3394 | 0.3747 |
| Branch 2 | 0.2330 | 0.3044 | 0.2319 | 0.2193 | 0.3739 | 0.3494 |
| Branch 3 | N/A | 0.2966 | 0.2078 | 0.1957 | 0.3941 | 0.3257 |
| Branch 4 | N/A | 0.4449 | 0.2403 | 0.2377 | 0.3617 | 0.3012 |

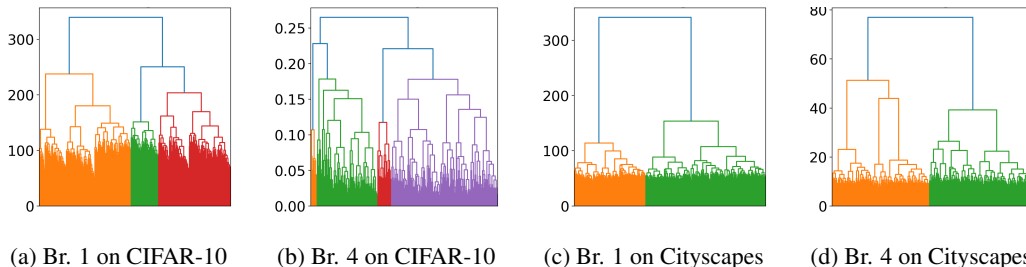

(a) Br. 1 on CIFAR-10     (b) Br. 4 on CIFAR-10     (c) Br. 1 on Cityscapes     (d) Br. 4 on Cityscapes

Figure 4: Dendrograms of intermediate embeddings of different branches on different datasets.

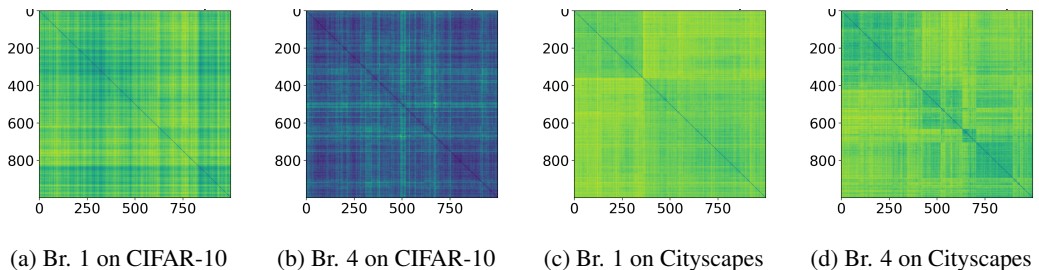

(a) Br. 1 on CIFAR-10     (b) Br. 4 on CIFAR-10     (c) Br. 1 on Cityscapes     (d) Br. 4 on Cityscapes

Figure 5: Distance heatmaps of intermediate embeddings of different branches on different datasets.

## 5 EXPERIMENT

### 5.1 HYPERBOLICITY

We evaluate the hyperbolicity of pre-trained MDEQ (Bai et al., 2020) models on CIFAR-10 (Krizhevsky et al., 2009), ImageNet (Russakovsky et al., 2015), and Cityscapes (Cordts et al., 2016). Relative hyperbolicity values $\delta_{rel}$ are reported in Table 1, where smaller values indicate stronger hyperbolicity and greater suitability for hyperbolic representations. Table 1 shows that, consistent with HyperbolicCV (Bdeir et al., 2024), branches 1 and 3 on CIFAR-10 and ImageNet are more hyperbolic than branches 2 and 4. Interestingly, all branches on Cityscapes are weakly hyperbolic, indicating that classification data encode clearer hierarchical structures than segmentation.

We further visualize embeddings via dendrograms (Figure 4), distance heatmaps (Figure 5), minimum spanning trees (MST) (Figure 6), and t-SNE (Figure 7). On CIFAR-10, Branch 1 consistently shows clearer hierarchy: dendrograms with long branches and distinct clusters, block-structured heatmaps, MST with extended backbones, and well-separated t-SNE clusters. Branch 4 is flatter, with diffuse heatmaps, collapsed MST, and entangled t-SNE distributions. On Cityscapes, all views are flat or uniform, confirming weak hierarchical separation. More visualizations in Appendix F.1.

These findings guide our design: for classification, we follow HyperbolicCV (Bdeir et al., 2024) by placing the head in hyperbolic space and adopting a hybrid encoder (branches 1 and 3 hyperbolic, branches 2 and 4 Euclidean). For segmentation, we find that a hyperbolic head matches Euclidean baselines, but extending hyperbolicity into the encoder degrades performance (Appendix F.3).

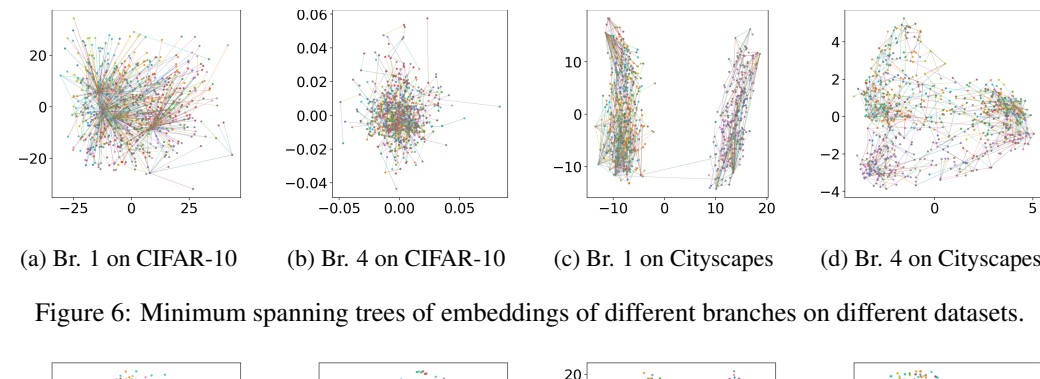

(a) Br. 1 on CIFAR-10    (b) Br. 4 on CIFAR-10    (c) Br. 1 on Cityscapes    (d) Br. 4 on Cityscapes

Figure 6: Minimum spanning trees of embeddings of different branches on different datasets.

(a) Br. 1 on CIFAR-10    (b) Br. 4 on CIFAR-10    (c) Br. 1 on Cityscapes    (d) Br. 4 on Cityscapes

Figure 7: t-SNE projections of intermediate embeddings of different branches on different datasets.

## 5.2 CLASSIFICATION ON CIFAR-10

Table 2 reports results on CIFAR-10. Without data augmentation, HIE-small surpasses both Euclidean baselines and hyperbolic counterparts, achieving the highest accuracy at comparable parameter counts. With standard augmentation, HIE attains 94.0%, outperforming MDEQ and all prior hyperbolic designs while maintaining the same model size. These results demonstrate that HIE consistently leverages hyperbolic geometry for stronger representation learning in low-data regimes and remains competitive with augmentation. Further details of the tasks, hyperparameters, and training settings are provided in Appendix E.

Table 2: Evaluation on CIFAR-10. Standard deviations are calculated on 5 runs.

| Model | Model Size | Accuracy (%) |
|---|---|---|
| CIFAR-10 (*without* data augmentation) | | |
| Neural ODEs (Chen et al., 2018) () | 172K | $53.7 \pm 0.2$ |
| Aug. Neural ODEs (Dupont et al., 2019) | 172K | $60.6 \pm 0.4$ |
| Single-stream DEQ (Bai et al., 2019) | 170K | $82.2 \pm 0.3$ |
| ResNet-18 (He et al., 2016) | 170K | $81.6 \pm 0.3$ |
| MDEQ-small (Bai et al., 2020) | 170K | $87.1 \pm 0.4$ |
| HECNN Lorentz (Bdeir et al., 2024) | 170K | $82.7 \pm 0.4$ |
| HCNN Lorentz (Bdeir et al., 2024) | 170K | $81.9 \pm 0.3$ |
| HIE-small (Ours) | 170K | $87.8 \pm 0.4$ |
| CIFAR-10 (*with* data augmentation) | | |
| ResNet-18 (He et al., 2016) | 10M | $92.9 \pm 0.2$ |
| MDEQ (Bai et al., 2020) | 10M | $93.8 \pm 0.3$ |
| HyperbolicNN (Ganea et al., 2018) | 10M | $88.8 \pm 0.5$ |
| Hybrid Poincaré (Guo et al., 2022) | 10M | $91.9 \pm 0.2$ |
| Hybrid Lorentz (Bdeir et al., 2024) | 10M | $92.7 \pm 0.2$ |
| Poincaré ResNet (Van Spengler et al., 2023) | 10M | $92.3 \pm 0.4$ |
| HECNN Lorentz (Bdeir et al., 2024) | 10M | $92.9 \pm 0.3$ |
| HCNN Lorentz (Bdeir et al., 2024) | 10M | $92.9 \pm 0.2$ |
| HIE (Ours) | 10M | $94.0 \pm 0.3$ |

## 5.3 CLASSIFICATION ON IMAGENET

Table 3 shows results on ImageNet. HIE-small achieves 75.7% accuracy with 18M parameters, improving upon both DEQ and hybrid hyperbolic baselines. Scaling to HIE-XL further increases performance to 79.3%, matching MDEQ-XL and avoiding out-of-memory failures that affect existing HyperbolicCV models at this scale. These results indicate that HIE scales effectively to large-scale classification, combining the scalability of implicit equilibrium models with the representational advantages of hyperbolic geometry. Further details of the experiments are provided in Appendix E.

## 5.4 EQUILIBRIUM CONVERGENCE

We compare the convergence of MDEQ and our proposed HIE on CIFAR-10, measuring residual change $\|z^{[i+1]} - z^{[i]}\|/\|z^{[i]}\|$ as in (Bai et al., 2020). Figure 8 shows averages over 10 runs with shaded deviations. Under Broyden's solver (a), HIE reaches equilibrium faster and more stably than Euclidean DEQ, while in forward unrolling (b), HIE continues to decay monotonically whereas DEQ stagnates. This demonstrates the curvature-induced contraction of hyperbolic operators, which accelerates and stabilizes equilibrium convergence.

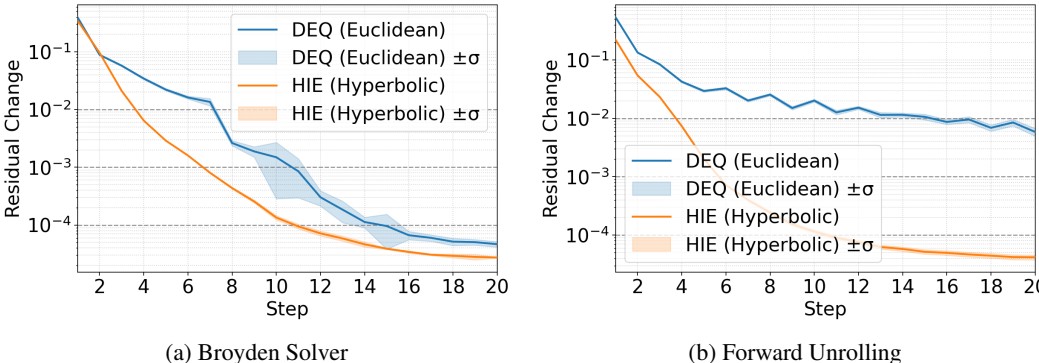

Figure 8: Convergence of MDEQ and HIE on CIFAR-10 over 10 runs. (a) Residual change when solving for the equilibrium with the Broyden solver. (b) Residual change when unrolling the network layer-by-layer without an equilibrium solver. Shaded regions denote $\pm$ one standard deviation.

## 5.5 MEMORY AND RUNTIME

We benchmark runtime and memory on ImageNet with various backbones and sizes (Table 4). Both MDEQ and our HIE scale to larger batches without out-of-memory errors, unlike HyperbolicCV (Bdeir et al., 2024). HIE further reduces runtime relative to Euclidean DEQs while keeping identical memory use. Results on CIFAR-10 are provided in Appendix F.2.

The gain follows from Section 5.4: hyperbolic operators contract more sharply than Euclidean ones, so fewer solver iterations are required to reach equilibrium. Thus HIE achieves faster inference with the $O(1)$ memory of implicit differentiation. In contrast, explicit hyperbolic designs require layer unrolling, and compounded geometric operations prevent state sharing, leading to rapid memory growth.

Table 3: Evaluation on ImageNet classification.

| Model | Model Size | Accuracy |
|---|---|---|
| AlexNet (Krizhevsky et al., 2012) | 238M | 57.0 |
| ResNet-18 (He et al., 2016) | 13M | 70.2 |
| ResNet-34 (He et al., 2016) | 21M | 74.8 |
| Inception-V2 (Ioffe & Szegedy, 2015) | 12M | 74.8 |
| ResNet-50 (He et al., 2016) | 26M | 75.1 |
| HRNet-W18-C (Wang et al., 2020) | 21M | 76.8 |
| Single-stream DEQ (Bai et al., 2019) | 18M | 72.9 |
| MDEQ-small (Bai et al., 2020) | 18M | 75.5 |
| HyperbolicNN (Ganea et al., 2018) | 13M | 65.7 |
| Hybrid Poincaré (Guo et al., 2022) | 13M | 68.9 |
| Hybrid Lorentz (Bdeir et al., 2024) | 13M | 70.1 |
| Poincaré ResNet (Van Spengler et al., 2023) | 10M | 67.0 |
| HECNN Lorentz (Bdeir et al., 2024) | 13M | 72.0 |
| HCNN Lorentz (Bdeir et al., 2024) | 13M | 71.7 |
| HECNN Lorentz (Bdeir et al., 2024) | 26M | 75.3 |
| HCNN Lorentz (Bdeir et al., 2024) | 26M | 75.1 |
| HIE (Ours) | 18M | 75.7 |
| ResNet-101 (He et al., 2016) | 52M | 77.1 |
| W-ResNet-50 (Zagoruyko & Komodakis, 2016) | 69M | 78.1 |
| DenseNet-264 (Huang et al., 2017) | 74M | 79.7 |
| MDEQ-large (Bai et al., 2020) | 63M | 77.5 |
| Unrolled 5-layer MDEQ-large (Bai et al., 2020) | 63M | 75.9 |
| MDEQ-XL (Bai et al., 2020) | 81M | 79.2 |
| HECNN Lorentz (Bdeir et al., 2024) | 81M | OOM |
| HCNN Lorentz (Bdeir et al., 2024) | 81M | OOM |
| HIE-XL (Ours) | 81M | 79.3 |

Table 4: Runtime and memory consumption on ImageNet.

| Model | Backbone | Model Size | Batch Size | Memory (GB) | Runtime (ms) |
|---|---|---|---|---|---|
| HECNN Lorentz (Bdeir et al., 2024) | ResNet-18 | 10M | 4 | 14.5 | 337 |
| HECNN Lorentz (Bdeir et al., 2024) | ResNet-18 | 10M | 8 | OOM | N/A |
| MDEQ (Bai et al., 2020) | MDEQ | 10M | 8 | 1.7 | 178 |
| HIE (Ours) | HIE | 10M | 8 | 1.7 | 142 |
| HECNN Lorentz (Bdeir et al., 2024) | ResNet-50 | 26M | 2 | 23.0 | 753 |
| HECNN Lorentz (Bdeir et al., 2024) | ResNet-50 | 26M | 4 | OOM | N/A |
| MDEQ-XL (Bai et al., 2020) | MDEQ | 81M | 4 | 2.1 | 31 |
| HIE-XL (Ours) | HIE | 81M | 4 | 2.1 | 28 |

## 6 CONCLUSION

We design *Hyperbolic Implicit Equilibrium* (HIE), a constant-memory framework for negatively curved spaces. Our theory provides convergence, stability, and generalization guarantees. HIE converges faster and more stably than DEQ while avoiding explicit hyperbolic memory blow-ups. It scales on CIFAR-10 and ImageNet with $O(1)$ memory, reduced runtime, and competitive accuracy.

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

## ETHICS STATEMENT

This work complies with the ICLR Code of Ethics. Our study uses only publicly available datasets (CIFAR-10 (Krizhevsky et al., 2009), ImageNet (Russakovsky et al., 2015), Cityscapes (Cordts et al., 2016)) and does not involve human subjects, private data, or sensitive content. We believe the broader impacts of this work are positive, as it advances efficient and scalable deep learning methods.

## REPRODUCIBILITY STATEMENT

We have made every effort to ensure reproducibility. All theoretical proofs are included in the Appendix D. Details of model architectures, training setups, and hyperparameters are provided in Sections 3, 5, and Appendix. Code and pretrained models will be released upon publication.

## THE USE OF LARGE LANGUAGE MODELS

We used a large language model (ChatGPT) solely to aid in language polishing and clarity improvements of the manuscript. The research ideas, method design, and experimental results are entirely our own. No part of the technical content or experimental findings is generated by an LLM.

## A    HYPERBOLIC GEOMETRY

This appendix summarizes the main operators in hyperbolic space, with a focus on the Lorentz model. Many of these constructions apply to general Riemannian manifolds, but here we emphasize the closed-form expressions that make the Lorentz model particularly attractive for deep learning applications (Cannon et al., 1997; Ratcliffe, 2006; Nickel & Kiela, 2018; Law et al., 2019; Chen et al., 2021).

## A.1 POINCARÉ BALL AND LORENTZ MODEL

**Poincaré Ball.** The $n$-dimensional Poincaré ball $\mathbb{B}_K^n = (\mathbb{B}^n, g^K)$ with curvature $K < 0$ is

$$\mathbb{B}^n = \left\{ x \in \mathbb{R}^n : -K\|x\|^2 < 1 \right\}, \quad g_x^K = \lambda_x^2 I, \quad \lambda_x = \tfrac{2}{1+K\|x\|^2}.$$

It describes hyperbolic space as an open ball of radius $\sqrt{-1/K}$.

**Lorentz Model.** The Lorentz model realizes $\mathbb{H}_K^d$ as the upper sheet of a two-sheeted hyperboloid embedded in $\mathbb{R}^{d+1}$ with bilinear form

$$\langle u, v \rangle_{\mathcal{L}} \;=\; -u_0 v_0 + \sum_{i=1}^{d} u_i v_i.$$

Thus,

$$\mathbb{H}_K^d \;=\; \{\, u \in \mathbb{R}^{d+1} : \langle u, u \rangle_{\mathcal{L}} = 1/K, \; u_0 > 0 \,\}.$$

## A.2 DISTANCE AND TANGENT SPACE

The geodesic distance between $x, y \in \mathbb{H}_K^d$ is

$$d_{\mathbb{H}_K}(x, y) = \tfrac{1}{\sqrt{-K}} \, \operatorname{arcosh}\big(K \langle x, y \rangle_{\mathcal{L}}\big). \tag{12}$$

An equivalent form for squared distance is (Law et al., 2019):

$$d_{\mathbb{H}_K}^2(x, y) \;=\; \tfrac{2}{K} - 2\langle x, y \rangle_{\mathcal{L}}.$$

The tangent space at $x \in \mathbb{H}_K^d$ is

$$T_x \mathbb{H}_K^d = \{v \in \mathbb{R}^{d+1} : \langle v, x \rangle_{\mathcal{L}} = 0\}.$$

## A.3 EXPONENTIAL AND LOGARITHMIC MAPS

For $z \in T_x \mathbb{H}_K^d$, define $\alpha = \sqrt{-K}\,\|z\|_{\mathcal{L}}$. The exponential map is

$$\exp_x^K(z) = \cosh(\alpha)\, x + \sinh(\alpha)\, \tfrac{z}{\alpha}. \tag{13}$$

Conversely, the logarithmic map for $y \in \mathbb{H}_K^d$ is

$$\log_x^K(y) = \frac{\operatorname{arcosh}(\beta)}{\sqrt{\beta^2 - 1}}\, (y - \beta x), \quad \beta = K\langle x, y \rangle_{\mathcal{L}}. \tag{14}$$

At the origin $o = [1/\sqrt{-K}, 0, \dots, 0]$, the exponential simplifies to

$$\exp_o^K(z) = \tfrac{1}{\sqrt{-K}}\big(\cosh(\sqrt{-K}\|z\|), \; \sinh(\sqrt{-K}\|z\|)\tfrac{z}{\|z\|}\big).$$

## A.4 PARALLEL TRANSPORT

The parallel transport of $v \in T_x \mathbb{H}_K^d$ along the geodesic from $x$ to $y$ is

$$\mathrm{PT}_{x \to y}^K(v) = v - \frac{\langle \log_x^K(y), v \rangle_{\mathcal{L}}}{d_{\mathbb{H}_K}(x, y)} \big(\log_x^K(y) + \log_y^K(x)\big) \tag{15}$$

$$= v + \frac{\langle y, v \rangle_{\mathcal{L}}}{1/(-K) - \langle x, y \rangle_{\mathcal{L}}}(x + y). \tag{16}$$

## A.5 LORENTZIAN CENTROID AND POOLING

The weighted Lorentzian centroid of $\{x_i\}_{i=1}^m$ with weights $\nu_i \geq 0$, $\sum_i \nu_i > 0$, solves $\min_\mu \sum_i \nu_i d_{\mathbb{H}_K}^2(x_i, \mu)$ and admits the closed form (Law et al., 2019):

$$\mu = \frac{\sum_i \nu_i x_i}{\sqrt{-K}\,\big\|\sum_i \nu_i x_i\big\|_{\mathcal{L}}}.$$

Average pooling in hyperbolic space can thus be implemented via the centroid over receptive fields.

## A.6 LORENTZ TRANSFORMATIONS

Lorentz transformations are linear maps $A \in \mathbb{R}^{(d+1) \times (d+1)}$ that preserve the bilinear form: $\langle Ax, Ay \rangle_{\mathcal{L}} = \langle x, y \rangle_{\mathcal{L}}$. They form the group $O^+(1, d)$. By polar decomposition, any such $A$ factors as a Lorentz rotation $R \in SO^+(1, d)$ and a Lorentz boost $B(v)$ determined by a velocity $v \in \mathbb{R}^d$, $\|v\| < 1$.

## A.7 LORENTZ FULLY-CONNECTED LAYER

Following Chen et al. (2021), linear maps in tangent space cannot realize all Lorentz transformations. Instead, one can directly parameterize in ambient space. Given $x \in \mathbb{H}_K^d$, weights $W \in \mathbb{R}^{m \times (d+1)}$, and bias $b$, define

$$y = \left[ \sqrt{\|\psi(Wx + b)\|^2 - 1/K}, \quad \psi(Wx + b) \right],$$

where $\psi$ includes nonlinearity and bias.

## A.8 CONCATENATION

Given points $\{x_i \in \mathbb{H}_K^d\}_{i=1}^N$, Lorentz direct concatenation produces a point $y \in \mathbb{H}_K^{Nd}$ by stacking their spatial components with an adjusted time coordinate to ensure validity on the hyperboloid.

## A.9 WRAPPED NORMAL DISTRIBUTION

A wrapped normal distribution on $\mathbb{H}_K^d$ can be constructed as (Nagano et al., 2019):

1. Sample $v \sim \mathcal{N}(0, \Sigma)$ in $\mathbb{R}^d$, extend to $[0, v] \in T_o \mathbb{H}_K^d$.
2. Parallel transport to $T_\mu \mathbb{H}_K^d$ for mean $\mu$.
3. Map to $\mathbb{H}_K^d$ via $\exp_\mu^K$.

This distribution has closed-form density and supports efficient sampling.

## A.10 MAPPING BETWEEN MODELS

The Lorentz and Poincaré models are isometric. A diffeomorphism mapping $x = [x_t, x_s] \in \mathbb{H}_K^d$ to the Poincaré ball is

$$p_{\mathbb{H} \to \mathbb{B}}(x) = \frac{x_s}{x_t + 1/\sqrt{-K}}.$$

# B HYPERBOLIC NEURAL NETWORKS

## B.1 COMPUTATION OF $\delta$-HYPERBOLICITY

Let $(\mathcal{X}, d)$ be a metric space. For any three points $\mathbf{u}, \mathbf{v}, \mathbf{w} \in \mathcal{X}$, the Gromov product with basepoint $\mathbf{x}$ is defined as

$$(\mathbf{v}, \mathbf{w})_{\mathbf{u}} = \tfrac{1}{2}\big(d(\mathbf{u}, \mathbf{v}) + d(\mathbf{u}, \mathbf{w}) - d(\mathbf{v}, \mathbf{w})\big). \tag{17}$$

The space is said to be $\delta$-hyperbolic if for all $\mathbf{u}, \mathbf{v}, \mathbf{w}, \mathbf{x} \in \mathcal{X}$,

$$(\mathbf{u}, \mathbf{w})_{\mathbf{x}} \geq \min\{(\mathbf{u}, \mathbf{v})_{\mathbf{x}}, (\mathbf{v}, \mathbf{w})_{\mathbf{x}}\} - \delta. \tag{18}$$

The smallest $\delta$ satisfying this inequality serves as the hyperbolicity constant of the space. For computational purposes, we approximate $\delta$ using the efficient min–max matrix product approach (Fournier et al., 2015), and normalize it by the dataset diameter to obtain a scale-invariant measure:

$$\delta_{\text{rel}}(\mathcal{X}) = \frac{2\delta(\mathcal{X})}{\text{diam}(\mathcal{X})}, \quad \text{diam}(\mathcal{X}) = \max_{\mathbf{u}, \mathbf{v} \in \mathcal{X}} d(\mathbf{u}, \mathbf{v}). \tag{19}$$

By construction, $\delta_{\text{rel}} \in [0, 1]$, where values closer to 0 indicate stronger hyperbolicity. This normalized score is used throughout our experiments to decide whether to employ hyperbolic or Euclidean components within the implicit equilibrium solver.

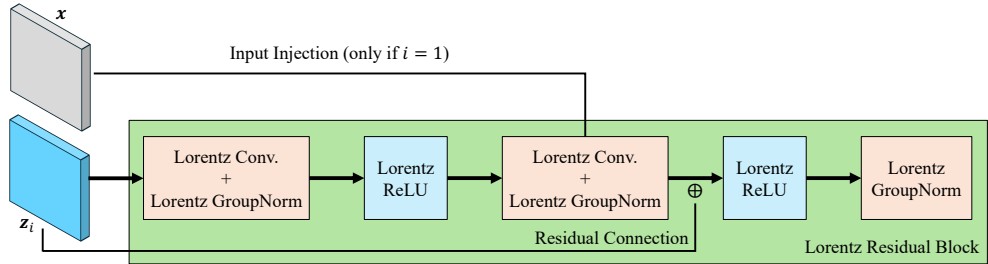

Figure 9: The Lorentz residual block used in HIE. An HIE contains only *one* such layer.

## B.2 HYPERBOLIC MODULES

The internal structure of the Lorentz residual block is shown in Figure 9. We use curvature $K < 0$ and set $K = -1$ for all experiments. Points on the manifold are in bold (e.g., $\mathbf{u} \in \mathbb{H}_K^d$). We denote the Lorentz inner product by $\langle \cdot, \cdot \rangle_{\mathcal{L}}$ and use the origin $\mathbf{o} = [1/\sqrt{-K}, \mathbf{0}]^\top$.

**Lorentz Convolutional Layer.** A hyperbolic feature map is an ordered grid of manifold points $\{\mathbf{u}_{h,w} \in \mathbb{H}_K^d\}_{h,w}$, stored in channel-last format by appending the time component as an extra channel. Given a kernel with receptive field size $\tilde{H} \times \tilde{W}$, we (i) collect the points in the local window, (ii) concatenate them hyperbolically, and (iii) apply a Lorentz fully-connected (LFC) transformation to obtain the output point:

$$\mathbf{y}_{h,w} \;=\; \mathrm{LFC}\Big(\mathrm{HCat}\big(\{\mathbf{u}_{h+\hat{h},\, w+\hat{w}}\}\big)\Big), \tag{20}$$

where HCat denotes Lorentz direct concatenation and LFC is a learnable Lorentz transformation that maps $\mathbb{H}_K^{d\tilde{H}\tilde{W}} \to \mathbb{H}_K^{d'}$. Concretely, letting $\psi(\cdot)$ be an element-wise nonlinearity in the *space* component,[1] the LFC can be implemented in ambient coordinates as

$$\mathbf{y} = \begin{bmatrix} \sqrt{\|\psi(W\,\mathbf{x}_s + \mathbf{b})\|_2^2 - 1/K} \\ \psi(W\,\mathbf{x}_s + \mathbf{b}) \end{bmatrix}, \quad \mathbf{x} = \begin{bmatrix} x_t \\ \mathbf{x}_s \end{bmatrix} \in \mathbb{H}_K^D, \tag{21}$$

with $W, \mathbf{b}$ learnable. Padding uses the origin $\mathbf{o}$ (hyperbolic "zero"). The transposed Lorentz convolution reuses the same operator with modified connectivity (origin-insertion upsampling).

**Lorentz MLR Classifier.** We model class regions via hyperbolic hyperplanes. Using the reparameterization by a direction $\mathbf{z} \in \mathbb{R}^d$ and an offset $a \in \mathbb{R}$, the hyperplane is

$$\widetilde{\mathcal{H}}_{\mathbf{z},a} = \Big\{ \mathbf{x} \in \mathbb{H}_K^d \;\Big|\; \cosh(\sqrt{-K}\,a)\,\langle \mathbf{z}, \mathbf{x}_s \rangle - \sinh(\sqrt{-K}\,a)\,\|\mathbf{z}\|_2\,x_t \;=\; 0 \Big\}, \tag{22}$$

where $\mathbf{x} = [x_t, \mathbf{x}_s]^\top$. The (signed) distance from $\mathbf{x}$ to $\widetilde{\mathcal{H}}_{\mathbf{z},a}$ is

$$d_{\mathbb{H}_K}(\mathbf{x}, \widetilde{\mathcal{H}}_{\mathbf{z},a}) = \frac{1}{\sqrt{-K}}\,\mathrm{asinh}\bigg( \frac{\sqrt{-K}\,\alpha}{\beta} \bigg),$$

$$\alpha = \cosh(\sqrt{-K}\,a)\,\langle \mathbf{z}, \mathbf{x}_s \rangle - \sinh(\sqrt{-K}\,a)\,\|\mathbf{z}\|_2\,x_t, \tag{23}$$

$$\beta = \sqrt{\|\cosh(\sqrt{-K}\,a)\,\mathbf{z}\|_2^2 - \big(\sinh(\sqrt{-K}\,a)\,\|\mathbf{z}\|_2\big)^2}.$$

For $C$ classes with parameters $\{(\mathbf{z}_c, a_c)\}_{c=1}^C$, the Lorentz MLR logits are proportional to (signed) distances:

$$\ell_c(\mathbf{x}) = \frac{1}{\sqrt{-K}}\,\mathrm{sign}(\alpha_c)\,\beta_c\,\Big| \mathrm{asinh}\big(\tfrac{\sqrt{-K}\,\alpha_c}{\beta_c}\big)\Big|, \quad \alpha_c, \beta_c \text{ as in equation 23 for } (\mathbf{z}_c, a_c). \tag{24}$$

Class probabilities follow by softmax over $\{\ell_c\}_{c=1}^C$.

---

[1] We avoid internal tangent-space shuttling to improve stability.

**Lorentz Residual Connection and Activation.** Residual addition is ill-defined on $\mathbb{H}_K^d$. We therefore perform *space-component addition* and recover the time component geometrically. Let the block input be $\mathbf{u} = [u_t, \mathbf{u}_s]^\top$ and the block transform output (in ambient coords) be $\mathbf{r} = [r_t, \mathbf{r}_s]^\top$. The residual output is

$$\mathbf{v} = \begin{bmatrix} \sqrt{\|\mathbf{u}_s + \mathbf{r}_s\|_2^2 - 1/K} \\ \mathbf{u}_s + \mathbf{r}_s \end{bmatrix} \in \mathbb{H}_K^d. \tag{25}$$

Non-linear activations act on the space component only; e.g., a Lorentz ReLU is

$$\mathrm{LReLU}\left(\begin{bmatrix} x_t \\ \mathbf{x}_s \end{bmatrix}\right) = \begin{bmatrix} \sqrt{\|\mathrm{ReLU}(\mathbf{x}_s)\|_2^2 - 1/K} \\ \mathrm{ReLU}(\mathbf{x}_s) \end{bmatrix}. \tag{26}$$

This avoids frequent $\log / \exp$ mappings and has shown better stability and efficiency than tangent-space activations in our setting.

### B.3 Lorentz Group Nomalization

Design notes and variants are as follows:

- **Batch-independence.** LGN mirrors Euclidean GroupNorm by using *within-sample, within-group* statistics $(\boldsymbol{\mu}_g, \sigma_g^2)$ only, where $\sigma_g^2 = \frac{1}{N_g} \sum_j d_{\mathbb{H}_K}(\mathbf{u}_j, \boldsymbol{\mu}_g)^2$. This avoids batch-coupled population estimates that are known to destabilize implicit fixed-point solvers in equilibrium models.

- **Choice of groups.** We form groups over *sets of Lorentz points* (e.g., disjoint subsets of spatial sites) rather than slicing coordinates of a single point, preserving the integrity of each manifold element. In practice we adopt a uniform partition of spatial locations into $G$ groups per resolution (other partitions are admissible as long as each site is treated as a whole Lorentz point).

- **Recovering special cases.** Setting $G{=}1$ yields a Lorentz *Instance* Normalization (per sample, per resolution). Replacing groups by the full mini-batch recovers *Lorentz Batch Normalization (LBN)*. While the latter is geometrically well-defined, it reintroduces batch dependence and is usually unfavorable for equilibrium solvers; we therefore default to group-wise, batch-independent LGN.

- **Numerical stability.** We use a small $\varepsilon > 0$, clamp arguments of $\mathrm{arcosh}(\cdot)$ to $[1{+}\tau, \infty)$ with $\tau \approx 10^{-6}$, and rely on closed-form Lorentz operators for general $K{<}0$ (Appendix A). In all experiments we set $K = -1$; for general $K$ the tangent norms and distances are scaled by $1/\sqrt{|K|}$, and all maps $(\exp^K, \log^K, \mathrm{PT}^K)$ are implemented in a $K$-aware manner. We also adopt channel-last storage so that each spatial site corresponds to one Lorentz point.

## C Multiscale Implicit Equilibrium

**Setup.** Let $f_\theta(z; x)$ be the transformation defining the (multiscale) dynamics, and introduce $g_\theta(z; x) := f_\theta(z; x) - z$. An equilibrium $z^\star$ is any solution of $g_\theta(z^\star; x) = 0$, equivalently $z^\star = f_\theta(z^\star; x)$, which we obtain by root finding on $g_\theta$ rather than by explicitly unrolling layers to depth $L$.

### C.1 Forward Pass (root finding)

The forward pass seeks $z^\star = \mathrm{Rootfind}(g_\theta; x)$ using a black-box solver (e.g., Newton or quasi-Newton). In practice we use (limited-memory) Broyden:

$$z^{(k+1)} = z^{(k)} - \alpha B^{(k)} g_\theta\left(z^{(k)}; x\right),$$

where $B^{(k)} \approx (Jg_\theta|_{z^{(k)}})^{-1}$ is updated from the most recent low-rank corrections; storing only the latest $m$ updates yields an L-Broyden scheme suitable for high-dimensional vision features.

**Multiscale state.** For MDEQ, the hidden state is a tuple $z = [z_1, \ldots, z_n]$ with different spatial resolutions and channel dimensions; we initialize $z_i^{(0)} {=} 0$ and solve for all scales *jointly* so that the

solver enforces cross-scale consistency at equilibrium. Limited-memory updates are crucial here due to the very large Jacobians encountered at realistic image resolutions.

**Stopping and memory.** We stop when $\|g_\theta(z^{(k)}; x)\|$ falls below a tolerance or when a cap on function evaluations is reached. Because the backward pass does not replay the forward trajectory, the training memory scales with the size of a *single* block rather than the (implicit) depth.

### C.2 BACKWARD PASS (IMPLICIT DIFFERENTIATION)

Given a loss $\ell = L(z^\star, y)$, the implicit function theorem yields gradient expressions that depend only on quantities at $z^\star$; we do *not* differentiate through the root-finding iterations. Writing $J_g = Jg_\theta |_{z^\star}$, the adjoint $\bar{z}$ is obtained by solving the linear system

$$\left(J_g^\top\right) \bar{z} = \frac{\partial \ell}{\partial z^\star}, \quad \text{equivalently} \quad \bar{z}^\top J_g + \frac{\partial \ell}{\partial z^\star} = 0,$$

which is implemented as a vector–Jacobian product (VJP) solve rather than forming $J_g$ explicitly. Once $\bar{z}$ is computed, parameter/input gradients follow from

$$\frac{\partial \ell}{\partial \theta} = \bar{z}^\top \frac{\partial f_\theta(z^\star; x)}{\partial \theta}, \qquad \frac{\partial \ell}{\partial x} = \bar{z}^\top \frac{\partial f_\theta(z^\star; x)}{\partial x},$$

with the minus sign absorbed by solving against $J_g^\top$ as above:contentReference[oaicite:9]index=9. These expressions coincide with the familiar DEQ formulas and make the backward pass depend on a *single* linear solve at the equilibrium, decoupled from the forward solver's trajectory.

**Remarks for multiscale equilibria.** The adjoint is defined over the concatenated (but dimensionally heterogeneous) state $z = [z_1, \ldots, z_n]$, and the VJP solve couples all resolutions, mirroring the forward fusion. In practice, the same limited-memory strategy used in the forward pass is applied to the backward linear solve to control memory and runtime at megapixel scales.

**Summary.** The forward computes $z^\star$ by root finding on $g_\theta$; the backward computes an adjoint $\bar{z}$ via one linear VJP solve and then applies local derivatives of $f_\theta$ at $z^\star$. This yields constant training memory and avoids backpropagating through the unrolled fixed-point iterations.

## D THEORETICAL PROOFS

**Preliminaries.** We work on $\mathbb{H}_K^d$ with $K < 0$, $\kappa = \sqrt{-K}$. Riemannian metrics at $z$ induce norms $\|\cdot\|_{g_z}$ on $T_z\mathbb{H}_K^d$. When needed, parallel transport identifies tangent vectors across nearby points.

### D.1 PROOF OF LEMMA 1

In geodesic polar coordinates at $\beta$, the differential of $\exp_\beta^K$ at $v$ with $r = \|v\|_{g_\beta}$ acts as identity along the radial direction and as the linear map scaling tangential directions by $\frac{\sinh(\kappa r)}{\kappa r}$, derived from the Jacobi field $J$ solving $J'' + \kappa^2 J = 0$ with $J(0) = 0$, $J'(0) = I$. Hence the operator norm equals $\sinh(\kappa r)/(\kappa r)$ for $\|v\| = r$. Invertibility of $\exp_\beta$ on $B_R(\beta)$ yields $d\log_\beta = (d\exp_\beta)^{-1}$ at corresponding points, so $\|d\log_\beta\| = \kappa r / \sinh(\kappa r)$. Taking suprema over $r \leq R$ gives the claimed bounds and $L_{\exp}(R)L_{\log}(R) = 1$.

### D.2 PROOF OF THEOREM 1

Let $z_i \in B_R(\beta)$, $u_i = \log_\beta(z_i)$ and $\Phi(u) = \tau A(S(u)) + (1-\tau)u + b(x)$. By Lemma 1,

$$d_{\mathbb{H}_K}(F(z_1), F(z_2)) \leq \|d\exp_\beta\|_{\sup,R} \cdot \|\Phi(u_1) - \Phi(u_2)\| = L_{\exp}(R)\|\Phi(u_1) - \Phi(u_2)\|.$$

Using $\|A\| \leq \alpha$ and $S$ being $\eta$-Lipschitz on $\{\|u\| \leq R\}$,

$$\|\Phi(u_1) - \Phi(u_2)\| \leq \tau\|A\| \cdot \|S(u_1) - S(u_2)\| + (1-\tau)\|u_1 - u_2\| \leq (\tau\alpha\eta + (1-\tau))\|u_1 - u_2\|.$$

Again by Lemma 1, $\|u_1 - u_2\| \leq L_{\log}(R) d_{\mathbb{H}_K}(z_1, z_2)$. Combining the inequalities yields

$$d_{\mathbb{H}_K}(F(z_1), F(z_2)) \leq L_{\exp}(R)(\tau\alpha\eta + (1-\tau))L_{\log}(R) d_{\mathbb{H}_K}(z_1, z_2) = \rho_{\mathbb{H}} d_{\mathbb{H}_K}(z_1, z_2).$$

Since $\rho_{\mathbb{H}} < 1$, $F$ is a contraction, so it admits a unique fixed point in $B_R(\beta)$ and the Picard iteration converges linearly with rate at most $\rho_{\mathbb{H}}$.

Table 5: Settings & hyperparameters of each task. "cls." means classification task, and "seg." means segmentation task. These models correspond to the ones reported in Tables 2, 3, and 7.

| | CIFAR-10 (cls.) | | ImageNet (cls.) | | Cityscapes (seg.) | |
| --- | --- | --- | --- | --- | --- | --- |
| | HIE-Small | HIE | HIE-Small | HIE-Large | HIE-Small | HIE-Large |
| Input Image Size | $32 \times 32$ | | $224 \times 224$ | | $1024 \times 512$ (train) | $2048 \times 1024$ (test) |
| Number of Epochs | 50 | 200 | 100 | 100 | 480 | 480 |
| Batch Size | 128 | 128 | 128 | 128 | 12 | 12 |
| Optimizer | Adam | Adam | SGD | SGD | SGD | SGD |
| (Start) Learning Rate | 0.001 | 0.001 | 0.05 | 0.05 | 0.01 | 0.01 |
| Nesterov Momentum | - | - | 0.9 | 0.9 | - | - |
| Weight Decay | 0 | 0 | 5e−5 | 1e−4 | 2e−4 | 3e−4 |
| Use Pre-trained Weights | - | - | - | - | Yes, from ImageNet | Yes, from ImageNet |
| Number of Scales | 3 | 4 | 4 | 4 | (Exact same model as in ImageNet) | |
| # of Channels for Each Scale | [8,16,32] | [28,56,112,224] | [32,64,128,256] | [80,160,320,640] | – | |
| Width Expansion (in the residual block) | 5× | 5× | 5× | 5× | – | |
| Normalization (# of groups) | GroupNorm(4) | GroupNorm(4) | GroupNorm(4) | GroupNorm(4) | – | |
| Weight Normalization | ✓ | ✓ | ✓ | ✓ | – | |
| # of Downsamplings Before Equilibrium Solver | 0 | 0 | 2 | 2 | – | |
| Forward Quasi-Newton Threshold $T_f$ | 15 | 15 | 22 | 22 | 27 | 27 |
| Backward Quasi-Newton Threshold $T_b$ | 18 | 18 | 25 | 25 | 30 | 30 |
| Limited-Mem. Broyden's Method Storage Size $m$ | 12 | 12 | 18 | 18 | 18 | 18 |
| Variational Dropout Rate | 0.2 | 0.25 | 0.0 | 0.0 | 0.03 | 0.05 |

## D.3 PROOF OF COROLLARY 1

For $F_{\mathbb{R}}(u) = \tau A u + (1 - \tau)u + b$, $\mathrm{Lip}(F_{\mathbb{R}}) = \tau\alpha + (1 - \tau) = \rho_{\mathbb{R}}$. By Theorem 1, $\rho_{\mathbb{H}} = \tau\alpha\eta + (1 - \tau) \leq \rho_{\mathbb{R}}$ and $<$ holds if $\tau\alpha > 0$ and $\eta < 1$.

## D.4 PROOF OF THEOREM 2

Consider $G(\theta, z) := F_\theta(z; x) - z$ so that $G(\theta, z^\star) = 0$. Differentiate w.r.t. $\theta$:

$$\partial_\theta G + D_z G\, \partial_\theta z^\star = 0 \quad \Rightarrow \quad (I - D_z F_\theta(z^\star; x))\, \partial_\theta z^\star = \partial_\theta F_\theta(z^\star; x).$$

Thus $\partial_\theta z^\star = (I - D_z F_\theta)^{-1} \partial_\theta F_\theta$. For any differentiable $\mathcal{L}$, $\nabla_\theta \mathcal{L} = (D_z \mathcal{L}(z^\star))\, \partial_\theta z^\star$. Taking norms in $(T_{z^\star}\mathbb{H}_K^d, g_{z^\star})$ and using submultiplicativity,

$$\|\nabla_\theta \mathcal{L}\| \leq \|(I - D_z F_\theta)^{-1}\|\, \|\partial_\theta F_\theta(z^\star; x)\|\, \|\nabla_z \mathcal{L}(z^\star)\|.$$

Since $\|D_z F_\theta(z^\star; x)\| \leq \rho_{\mathbb{H}} < 1$, the Neumann series gives $\|(I - D_z F_\theta)^{-1}\| \leq 1/(1 - \rho_{\mathbb{H}})$, proving equation 10.

## D.5 PROOF OF LEMMA 2

In the Lorentz model, for unit time-like vectors $x, y$ (Lorentz norm $-1$), the geodesic distance satisfies $\cosh(\kappa d_{\mathbb{H}_K}(x, y)) = -\langle x, y\rangle_{\mathrm{Lor}}$. Writing $x = \exp_\beta(v)$, $y = \exp_\beta(w)$ and expressing $v, w$ in geodesic polar coordinates gives the stated hyperbolic law of cosines; see standard Riemannian geometry texts.

## D.6 PROOF OF THEOREM 3

With $r_1 = r_2 = r$ in Lemma 2, and $\cos\theta = 1 - 2\sin^2(\theta/2)$,

$$\cosh(\kappa\gamma_{\mathbb{H}}) = 1 + 2\sinh^2(\kappa r)\sin^2\left(\tfrac{\theta_0}{2}\right) = \cosh\left(2\,\mathrm{arcsinh}\left(\sinh(\kappa r)\sin\tfrac{\theta_0}{2}\right)\right).$$

Taking arcosh yields $\gamma_{\mathbb{H}} = \frac{2}{\kappa}\mathrm{arcsinh}(\sinh(\kappa r)\sin(\theta_0/2))$. Since $\sinh(\kappa r) \geq \kappa r$ and arcsinh is increasing and concave on $[0, \infty)$, for $s = \sin(\theta_0/2) \in [0, 1]$,

$$\mathrm{arcsinh}(\sinh(\kappa r)s) \geq s\,\mathrm{arcsinh}(\sinh(\kappa r)) = s\,\kappa r,$$

hence $\gamma_{\mathbb{H}} \geq 2rs = \gamma_{\mathbb{R}}$, with strict inequality when $r > 0$ and $\theta_0 \in (0, \pi)$.

## E ADDITIONAL IMPLEMENTATION DETAILS

We provide the implementation details in Table 5.

# F ADDITIONAL EXPERIMENTAL RESULTS

## F.1 HYPERBOLICITY

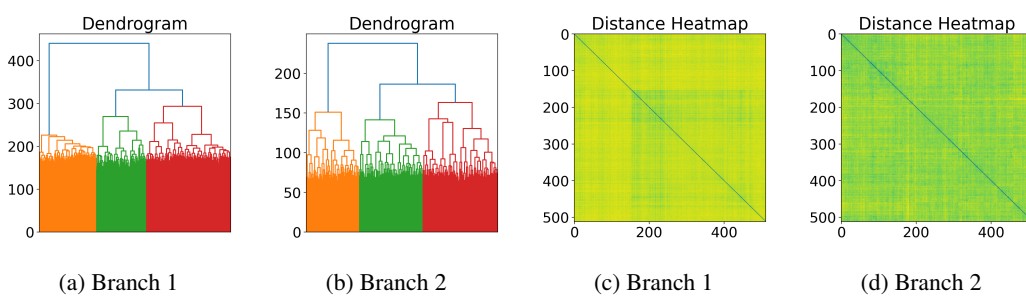

(a) Branch 1     (b) Branch 2     (c) Branch 1     (d) Branch 2

Figure 10: MDEQ-small on CIFAR-10.

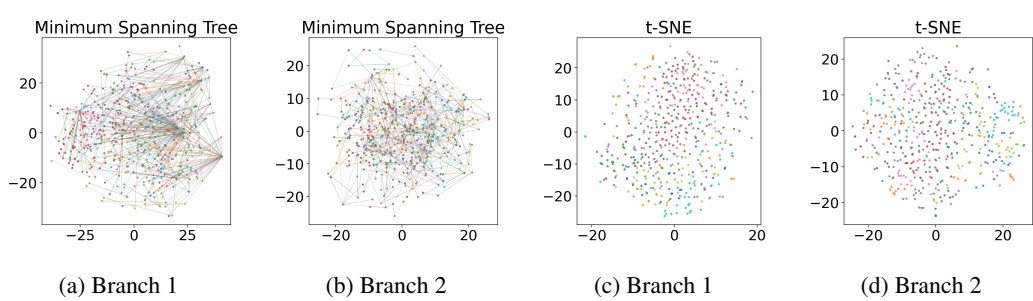

(a) Branch 1     (b) Branch 2     (c) Branch 1     (d) Branch 2

Figure 11: MDEQ-small on CIFAR-10.

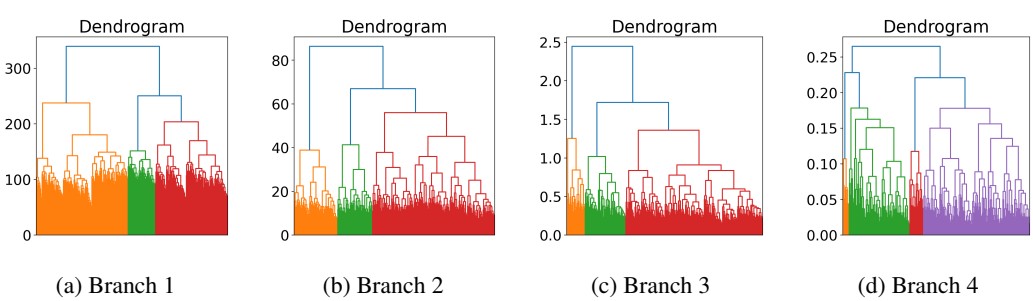

(a) Branch 1     (b) Branch 2     (c) Branch 3     (d) Branch 4

Figure 12: MDEQ-large on CIFAR-10.

## F.2 RUNTIME

See Table 6.

## F.3 SEMANTIC SEGMENTATION

See Table 7.

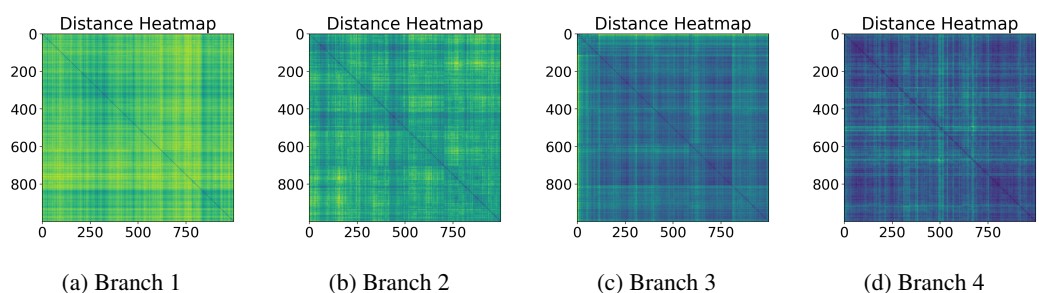

(a) Branch 1     (b) Branch 2     (c) Branch 3     (d) Branch 4

Figure 13: MDEQ-large on CIFAR-10.

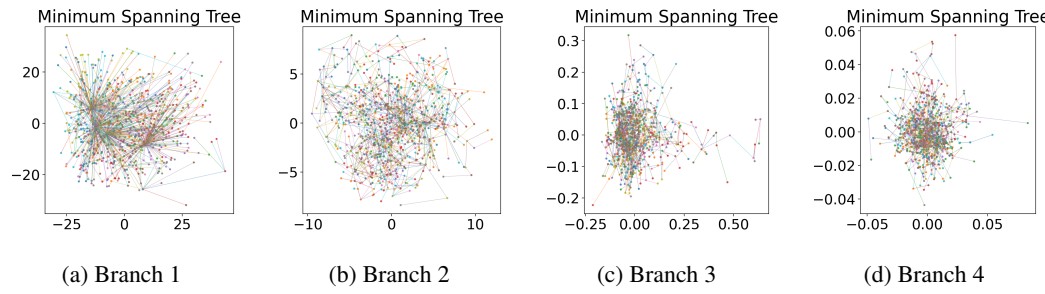

(a) Branch 1     (b) Branch 2     (c) Branch 3     (d) Branch 4

Figure 14: MDEQ-large on CIFAR-10.

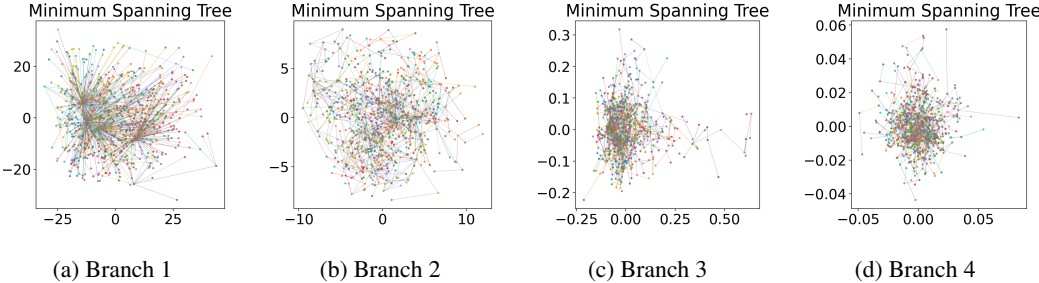

(a) Branch 1     (b) Branch 2     (c) Branch 3     (d) Branch 4

Figure 15: MDEQ-large on CIFAR-10.

Table 6: Runtime and memory consumption on CIFAR-10 (benchmarked on input batch size 32).

| Model | Backbone | Model Size | Memory (GB) | Runtime (ms) |
|---|---|---|---|---|
| HECNN Lorentz (Bdeir et al., 2024) | ReNet-18 | 10M | 2.8 | 206 |
| MDEQ (Bai et al., 2020) | MDEQ | 10M | 0.7 | 61 |
| HIE (Ours) | HIE | 10M | 0.7 | 43 |
| HECNN Lorentz (Bdeir et al., 2024) | ReNet-50 | 26M | 8.3 | 596 |
| HECNN Lorentz (Bdeir et al., 2024) | ReNet-101 | 52M | OOM | N/A |
| MDEQ-XL (Bai et al., 2020) | MDEQ | 81M | 1.5 | 28 |
| HIE-XL (Ours) | HIE | 81M | 1.5 | 24 |

Table 7: Evaluation on Cityscapes *val* semantic segmentation. Higher mIoU is better.

| Method | Backbone | Model Size | mIoU |
|---|---|---|---|
| ResNet-18-A (Liu et al., 2019) | ResNet-18 | 3.8M | 55.4 |
| ResNet-18-B (Liu et al., 2019) | ResNet-18 | 15.24M | 69.1 |
| MobileNetV2Plus (Sandler et al., 2018) | MobileNetV2 | 8.3M | 74.5 |
| GSCNN (Takikawa et al., 2019) | ResNet-50 | - | 73.0 |
| HRNetV2-W18-Small-v2 (Wang et al., 2020) | HRNet | 4.0M | 76.0 |
| MDEQ-small (Bai et al., 2020) | MDEQ | 7.8M | 75.1 |
| HIE-small (ours) | HIE | 7.8M | 75.3 |
| U-Net++ (Zhou et al., 2018b) | ResNet-101 | 59.5M | 75.5 |
| Dilated-ResNet (Yu et al., 2017) | D-ResNet-101 | 52.1M | 75.7 |
| PSPNet (Zhao et al., 2017) | D-ResNet-101 | 65.9M | 78.4 |
| DeepLabv3 (Chen et al., 2017) | D-ResNet-101 | 58.0M | 78.5 |
| PSANet (Zhao et al., 2018) | ResNet-101 | - | 78.6 |
| HRNetV2-W48 (Wang et al., 2020) | HRNet | 65.9M | 81.1 |
| MDEQ-large (Bai et al., 2020) | MDEQ | 53.0M | 77.8 |
| MDEQ-XL (Bai et al., 2020) | MDEQ | 70.9M | 80.3 |
| HIE-XL (ours) | HIE | 70.9M | 80.0 |

