# OpenReview forum: "Hyperbolic Implicit Equilibrium"
_ICLR.cc/2026/Conference — Submitted to ICLR 2026_

### Official Review · Reviewer_qoZ6 · 2025-10-22

**Soundness:** 2
**Presentation:** 1
**Contribution:** 2
**Rating:** 2
**Confidence:** 3

**Summary:**

This paper introduces a hyperbolic version of multiscale deep equilibrium models alongside some new operations such as the Lorentz group normalization. Their proposed method is tested on several vision datasets and its efficiency is compared in terms of memory usage and runtime.

**Strengths:**

- The paper contains a lot of interesting visualizations of the embeddings of MDEQ.
- The authors have included an extensive amount of baselines into their experiments.
- The proposed method appears to be significantly cheaper both in terms of memory and computation compared to the baselines.

**Weaknesses:**

The paper has several serious issues in my opinion, many of which seem to result from the writing, but because of which I cannot meaningfully judge the soundness of all parts of the proposed method:
- Many claims are made without proofs or references. For example, equation (4) mentions something apparently called the Lorentz mean, which the authors claim minimizes the Fréchet energy, which would implicate that it is the Fréchet mean. However, there is no citation and no proof of this actually being the Fréchet mean (which I am quite sure it is not). Furthermore, some Lorentz operations that the authors use are introduced in the appendix without reference and without motivation, while several of these operations (such as equations (21) and (25)) do not seem to depend on the metric of the hyperbolic space at all, in which case they cannot be dubbed hyperbolic operations in my opinion. There are several more of such claims, formulations etc. scattered throughout the paper.
- There are many citations missing in important places in general. For example, the introduction mentions HNN, but Ganea et al. aren't cited until the related work section.
- It is unclear over what the Lorentz Group Normalization operation actually groups, but it appears that the grouping is over the batch dimension. If this is the case, then it is not a hyperbolic version of group normalization (which groups over the channel dimension). It would simply be batch normalization applied to smaller groups within the batch. Furthermore, even if this removes batch size dependence, it introduces a new hyperparameter, which seems to defeat the purpose to me. Lastly, the naming of this operation is a bit unfortunate, since the Lorentz group is typically known as the group of all Lorentz transformations on Minkowski spacetime, which can lead to some confusion.
- The theory section is just a collection of statements. Without context, this is not useful to the reader. Moreover, some of the statements seem nonsensical. For example, Lemma 1 is about bi-Lipschitz bounds for the exponential and logarithmic maps, but provides only a lower bound for the former and an upper bound for the latter. Having only a lower bound does not prove that a function is Lipschitz, so I don't understand the naming of this theorem, nor its purpose. The proofs also appear overly complicated and hand-wavy to me. For example in the proof of Lemma 1, you can simply get these bounds by computing the derivatives normally, which is very straightforward.
- The implicit equilibrium appears to take hidden states of all resolutions as inputs, but it is unclear where these come from. I would expect that these result from some encoder. If this is the case, then you still have to backpropagate through the encoder to actually get meaningful input for the implicit network. I am not too familiar with implicit networks, so maybe the misunderstanding here is on me, but either way, I think the writing should be clearer on this part of the method.
- The paper estimates how hyperbolic some feature space is by computing the normalized delta hyperbolicity over a finite sample taken from the distribution over this feature space. Delta hyperbolicity tells us to what degree a metric space is hyperbolic. However, computing the delta hyperbolicity over a finite sample of a distribution over a continous space only tells us how hyperbolic this particular sample is and not necessarily whether the distribution from which the sample is taken is hyperbolic. For example, if we sample 4 perfectly colinear points from a uniform distribution over the Euclidean unit disk, then the delta hyperbolicity will tell us that it is perfectly hyperbolic, which it obviously is not. Whether the delta hyperbolicity of a sample is a good estimator of the delta hyperbolicity of its distribution is a very interesting question, but requires careful consideration (and I do not think that it actually holds). Therefore, the paper should include either a proof of this statement or a reference to a paper with such a proof in my opinion.
- Section 5.1 contains some analysis that is supposed to indicate how hyperbolic certain branches are. It is unclear to me what these branches are and the conclusions that are being drawn in lines 369-374 appear unfounded and subjective to me.
- Due to the lack of clarity in the earlier parts of the paper, I struggle to judge the results presented in sections 5.2, 5.3 and 5.5. However, I can say that reporting OOM seems unfair to me. Unless a single sample instantly causes OOM, it seems to me that higher batch sizes can be simulated through gradient accumulation.

In my opinion, the paper currently has too many problems that require significant rewriting as I think the method is currently not clear from the paper and its motivation is lacking.

**Questions:**

Could you address the concerns I raised under weaknesses and help clarify any points of confusion?

---

### Official Review · Reviewer_Tigq · 2025-10-27

**Soundness:** 3
**Presentation:** 2
**Contribution:** 2
**Rating:** 4
**Confidence:** 3

**Summary:**

This paper proposes hyperbolic implicit equilibrium (HIE), which combines hyperbolic neural networks (HNN) with deep equilibrium (DEQ) framework. In particular, HIE solves for a fixed-point representation in hyperbolic (Lorentz) space. The authors develop Lorentz group normalization to preserve manifold consistency. They also provide theoretical guarantees for convergence, stability, and bounded gradients, showing that negative curvature enhances contraction and accelerates equilibrium convergence. Experiments on CIFAR-10 and ImageNet demonstrate efficient and stable training.

**Strengths:**

1. This paper has a clear motivation to use an implicit equilibrium framework to address scalability issue of hyperbolic networks.
2. This paper has a full mathematical analysis that provides convergence, contraction, and margin guarantees.
3. The memory and runtime benchmarks are well presented and generally support the claim that the proposed HIE achieves improved efficiency.

**Weaknesses:**

1. Many components of the method are adaptations of the existing Euclidean DEQ framework, with limited novel algorithmic contributions. The work often feels like a Lorentzian reformulation of DEQ rather than a fundamentally new framework (as can be seen from Appendix C).
2. The paper’s experiments are restricted to CIFAR-10 and ImageNet, primarily using CNN/ResNet-style convolutional backbones. However, hyperbolic neural networks are traditionally applied to hierarchical or relational data (e.g., WordNet, knowledge graphs, or tree-structured text). It is improper to only consider image data. Also, baselines are limited. Interesting comparison should include (Shimizu et al., 2020, Chen et al., 2021) etc. as you cited in Section 2.1. Of course, these can be better evaluated for non-CV datasets.
3. The reported improvements in accuracy, memory, and runtime over Euclidean DEQs are tiny and not significant. It is unclear whether HIE really improves over MDEQ (Bai et al., 2020).

**Questions:**

Why do you only consider image data? Could you demonstrate the method’s effectiveness on other types of hierarchical benchmarks?

---

### Official Review · Reviewer_9zGu · 2025-10-31

**Soundness:** 3
**Presentation:** 2
**Contribution:** 2
**Rating:** 6
**Confidence:** 4

**Summary:**

This paper proposes a hyperbolic deep-equilibrium vision model (HIE) where the embbedding are hyperbolic and solved as a fixed point instead of stacking layers on top of each other, which is based on the MDEQ model. On top of the hyperbolic MDEQ architecture, the paper also proposes hyperbolic GroupNorm in Lorentz space. Theoretically, the paper proves that the hyperbolic MDEQ model contracts at the higher rate than Euclidean MDEQ, has stable gradient, and has better class separation. Experimentally, the paper shows that HIE achieves higher performance on image tasks and has overall lower runtime and faster convergence than Euclidean MDEQ.

**Strengths:**

1. The theoretical results for HIE shows better contraction factors than the Euclidean model, and this in term is experimentally validated to have a faster model convergence rate.
2. HIE achieves better performance on image tasks and faster runtime than the Euclidean MDEQ model

**Weaknesses:**

1. I find the presentation of Section 4 fairly confusing. There isn't enough discussion for the theoretical results to connect to the rest of the paper and the model being proposed
2. Also on Section 4, lemma 1 and lemma 2 are classic results for Riemannian and hyperbolic geometry. The current presentation make it seems as if the authors are claiming these are their own contributions. Additionally, theorems are usually reserved for major theoretical results and typically requires extensive proof components. The authors could reframe part of the sections with propositions instead.
3. The method is essentially a combination of MDEQ and existing hyperbolic components (GroupNorm is essentially reframing BatchNorm from Bdeir el al. 2024 and Van Spengler et al., 2023). Demonstrating effectiveness on additional experiments on top of just image classification would strengthen the paper's contribution.

**Questions:**

1. Table 4 shows HIE to have faster runtime than MDEQ. However, hyperbolic operations comes with added complexity, e.g. log and exp maps for GroupNorm. Could the authors provide additional justification on this?

---

### Official Review · Reviewer_ZZ3G · 2025-11-01

**Soundness:** 3
**Presentation:** 3
**Contribution:** 3
**Rating:** 4
**Confidence:** 4

**Summary:**

The paper introduces Hyperbolic Implicit Equilibrium (HIE), a novel framework that combines hyperbolic neural representations with implicit equilibrium models to enable deep, memory-efficient networks. By directly solving for fixed points using implicit differentiation, HIE overcomes the memory and runtime limitations of explicit hyperbolic architectures while leveraging the contraction properties of hyperbolic geometry for faster and more stable convergence. The framework incorporates a hybrid architecture, placing hyperbolic layers selectively based on layer-wise hyperbolicity, and introduces Lorentz Group Normalization to preserve manifold consistency. Theoretical analysis provides guarantees on convergence, stability, and generalization, while experiments on CIFAR-10, ImageNet, and Cityscapes demonstrate superior accuracy and efficiency compared to both Euclidean deep equilibrium models and prior hyperbolic networks. HIE highlights the potential of combining brain-inspired hyperbolic representations with implicit deep learning to scale hierarchical learning to larger and more complex tasks.

**Strengths:**

1. HIE allows effectively infinite-depth hyperbolic networks with constant memory usage, overcoming the memory and runtime bottlenecks of previous explicit hyperbolic networks.

2. Leveraging hyperbolic contraction, HIE converges to equilibrium faster and more stably than Euclidean DEQs, reducing solver iterations and runtime.

3. HIE Outperforms both Euclidean DEQs and prior hyperbolic networks on CIFAR-10 and ImageNet, achieving higher accuracy with competitive model sizes.

**Weaknesses:**

1. The performance of HIE depends on iterative equilibrium solvers (e.g., Broyden method), therefore the convergence is not guaranteed if assumptions about contraction are violated.

2. Incorporates multiple hyperbolic operations (exp/log maps, Lorentz group normalization, parallel transport), which may be harder to implement and more computationally expensive than standard Euclidean models.

3. While HIE scales better than prior hyperbolic models, runtime still grows with solver iterations and batch size.

4. Experiments on Cityscapes show weak hyperbolicity. As a consequence, extending hyperbolic layers in segmentation encoders can degrade performance, limiting generalization to some tasks.

**Questions:**

1. How does the selective placement of hyperbolic layers based on δ-hyperbolicity affect model performance on tasks with weakly hierarchical data, such as segmentation?

2. What are the limitations of HIE when scaling to extremely large models or datasets beyond those tested, and how might solver convergence or numerical stability be affected?

---

### Meta-Review · Area_Chair_mVeV · 2026-01-08

**Summary:**

This paper introduces Hyperbolic Implicit Equilibrium (HIE), combining hyperbolic representations with deep equilibrium models to achieve effectively infinite depth at constant memory cost. Reviewers agree the motivation is sound and the idea timely: negative curvature plausibly improves contraction and stability in implicit models. The work is technically competent and ambitious, but its execution is uneven.

Reviewers agreed that the paper is technically solid and tackles a timely question: How to scale hyperbolic neural networks using implicit equilibrium models.  However, several concerns tempered enthusiasm.  The most consistent issue is novelty and positioning: multiple reviewers view the method as a Lorentzian reformulation of existing deep equilibrium architectures rather than a fundamentally new framework, with some theoretical results presented as original despite being classical. Clarity and exposition were also recurring problems, particularly in the theory section, which reviewers found poorly motivated, insufficiently connected to the model, and lacking key citations. On the empirical side, evaluation is narrow, focusing almost exclusively on vision benchmarks, despite hyperbolic methods being most naturally motivated by hierarchical or relational data; reported gains over Euclidean DEQs are modest and sometimes fragile. Finally, several reviewers noted ambiguity in method definition, especially around Lorentz Group Normalization and the interpretation of hyperbolicity measurements.

There is no rebuttal.

For these reasons, the AC recommends rejection.  A substantially revised version, featuring clearer attribution, tighter theoretical exposition, and broader experimental support, could make a stronger case in a future submission.

**Reviewer Concerns:**

Reviewers agreed that the paper is technically solid and tackles a timely question: How to scale hyperbolic neural networks using implicit equilibrium models.  However, several concerns tempered enthusiasm.  The most consistent issue is novelty and positioning: multiple reviewers view the method as a Lorentzian reformulation of existing deep equilibrium architectures rather than a fundamentally new framework, with some theoretical results presented as original despite being classical. Clarity and exposition were also recurring problems, particularly in the theory section, which reviewers found poorly motivated, insufficiently connected to the model, and lacking key citations. On the empirical side, evaluation is narrow, focusing almost exclusively on vision benchmarks, despite hyperbolic methods being most naturally motivated by hierarchical or relational data; reported gains over Euclidean DEQs are modest and sometimes fragile. Finally, several reviewers noted ambiguity in method definition, especially around Lorentz Group Normalization and the interpretation of hyperbolicity measurements.

There is no rebuttal.

**Reviewer Scores:**

There is no rebuttal.

---

### Decision · Program_Chairs · 2026-01-26

Reject